# Prevalence and correlates of disability in Latin America and the Caribbean: Evidence from 8 national censuses

Samuel Berlinski[1,2]* , Suzanne Duryea[1], Santiago M. Perez-Vincent[1]

**1** Inter-American Development Bank, Washington, D.C., United States of America, **2** IZA Institute of Labor Economics, Bonn, Germany

☉ These authors contributed equally to this work.
* samuelb@iadb.org

**Data Availability Statement:** The data used in this study is owned by IPUMS International and cannot be shared by the authors. However, the data is publicly available and can be downloaded for free upon registration (https://international.ipums.org/

## Abstract

We estimate disability prevalence rates and gaps in social conditions in eight Latin America and the Caribbean (LAC) countries and project current and future disability prevalence rates in the region. Using data from representative samples of the population in eight countries, we find that reported disability prevalence varies widely across countries, ranging between 4.5 percent in Trinidad and Tobago (2011) to 24.9 percent in Brazil (2010). Differences in surveying approaches and demographic structures likely explain a part of this variation. We find marked sociodemographic gradients for disability. We also report significant disability gaps: people living with disabilities have lower educational attendance and completion rates and lower employment rates. We use age and sex-specific disability rates from our sample of countries and information on the current and future demographic structures in LAC countries to project disability prevalence for the whole region. We project that the total number of people with disabilities in this region will increase by approximately 60 million between 2020 and 2050. Our projections suggest that countries need to systematically plan and implement inclusion policies to adequately address the growing population of people with disabilities in the years to come.

## Introduction

A firm commitment to promote the social and economic inclusion of people with disabilities has emerged within Latin America and the Caribbean. As of 2020, all countries in the region have ratified the UN Convention on the Rights of Persons with Disabilities (CRPD). As governments have begun to implement the convention, there is an increasing need for policy-relevant indicators that involve the measurement of disability. The monitoring of the 2030 Sustainable Development Goals also requires high-quality disability data to assess progress toward those targets that explicitly disaggregate by disability status. These targets include eliminating gaps in access to education (Target 4.5), expanding employment opportunities (Target 8.5), and reducing the proportion of people with disabilities living below 50 percent of median income (Target 10.2). The COVID-19 crisis and its disproportionate impact on vulnerable

international/). The authors had no special access privileges to the data and other researchers will be able to access the data in the same manner as the authors. The authors have also provided a STATA code file that explains what variables to download from IPUMS International and the code to replicate all our tables and figures.

**Funding:** The 3 authors (Samuel Berlinski, Suzanne Dureya and Santiago Perez-Vincent) are employees of the Inter-American Development Bank. No specific grants were received to produce this document. The Inter-American Development Bank, its board of directors or the countries they represent had no role in study design, data collection and analysis, decision to publish, or preparation of the manuscript.

**Competing interests:** The authors have declared that no competing interests exist. Our affiliation to the Inter-American Development Bank does not alter our adherence to PLOS ONE policies on sharing data and materials.

populations, including people with disabilities, increase the challenge of achieving these goals. More than ever, this context demands careful planning, budgeting, monitoring, and evaluation of inclusion policies. These tasks require having credible measures and projections of people living with a disability.

Using data from eight of the most recent censuses in Latin America and the Caribbean, we estimate disability prevalence by age and sex circa 2010. Seven of the countries measured disability using a version of the United Nations' Washington Group on Disability Statistics (WG) questions that were formulated to measure disability consistent with the social model in the CRPD, although none strictly implemented the recommendations. We then examine the socioeconomic gradient of disability, complementing studies in the region [1] and globally [2] that highlight the importance of empirically documenting socioeconomic gaps. We also estimate gaps in education and employment by disability status, providing important baseline information on the socioeconomic inclusion of people with disabilities. Finally, using age and sex-specific disability rates from our sample of countries and information on the future demographic structures in each country, we estimate disability prevalence for 2020 and project it for 2050 for twenty-six countries in the region. The projections highlight the need for countries to systematically plan and implement inclusion policies to address the growing population of people with disabilities in the years to come.

Our paper contributes to the knowledge and capacity gap in documenting the situation of people with disabilities, as underscored in a recent UN report [3]. Whereas previous studies and reports [4, 5] have published estimates for broad age groups, this paper shows how large representative samples can be used to produce precise estimates for small sex-age groups that are well-suited to inform the regional discussion on inclusion policies. The paper also contributes to the set of studies that highlight how the approach (and the instrument) used to measure disability influences reported prevalence rates across high, middle, and low-income countries [6–9]. These findings are relevant to many countries in the region planning to collect their decennial censuses over the next two years, including Bolivia, Brazil, Chile, Ecuador, Panama, Paraguay, El Salvador, and thirteen countries in the Caribbean.

## Data and methods

### Data sources and definitions

We use data from eight of the most recent population censuses in Latin America and the Caribbean (LAC) with questions on individual disability status. Our inclusion criteria was the harmonized census microdata from the Integrated Public Use Microdata Series (IPUMS) International in which the questionnaires from the 2010 census round in LAC were available in Spanish or English and the disability data was available. Minnesota Population Center. Integrated Public Use Microdata Series, International: Version 7.1 [dataset]. Minneapolis, MN: IPUMS, 2018. https://doi.org/10.18128/D020.V7.1. These data come from censuses and surveys collected by National Statistics Offices in each country. Brazil: Institute of Geography and Statistics; Costa Rica: National Institute of Statistics and Censuses; Dominican Republic: National Statistics Office; Ecuador: National Institute of Statistics and Censuses; Mexico: National Institute of Statistics, Geography, and Informatics; Panama: Census and Statistics Directorate; Trinidad and Tobago: Central Statistical Office; Uruguay: National Institute of Statistics. The country (year) samples used are Brazil (2010), Costa Rica (2011), Dominican Republic (2010), Ecuador (2010), Mexico (2010), Panama (2010), Trinidad and Tobago (2011), and Uruguay (2011). The IPUMS data for the 2010 census of Argentina and the 2012 census of Bolivia do not include the disability variables and as such could not be used for the study. The countries cover a wide range of levels of economic development and age structures.

**Table 1. Sample size and characteristics.**

| Sample | Sample Size | | | Disability | | |
|---|---|---|---|---|---|---|
| | Pop. Fraction | # Persons | # Households | Approach | Questions/Categories | Missing |
| Brazil 2010 | 5% | 9,693,058 | 2,907,278 | F | 4 | 0.03% |
| Costa Rica 2011 | 10% | 430,082 | 124,693 | F | 7 | 0.00% |
| Dom. Republic 2010 | 10% | 943,784 | 268,637 | F | 12 | 1.31% |
| Ecuador 2010 | 10% | 1,448,233 | 386,944 | I | 5 | 0.63% |
| Mexico 2010 | 10% | 11,938,402 | 2,903,640 | F | 7 | 0.00% |
| Panama 2010 | 10% | 341,118 | 95,579 | F | 6 | 0.54% |
| Trinidad & Tob. 2011 | 8.80% | 116,917 | 35,824 | I | 6 | 8.11% |
| Uruguay 2011 | 10% | 328,425 | 118,498 | F | 4 | 3.55% |

Samples are provided by [10] from censuses and surveys collected by National Statistics Offices in each country. Approach: I stands for "impairment approach". F stands for "functional approach". Questions/Categories: number of different activities or physical conditions distinguished in the survey (counting "other" as one category). We code as *missing* all respondents for whom, based on available information, we cannot establish their disability status. The proportion of *missing* observations is computed considering individuals aged 3 and older. *In Uruguay 2011, children between 2 and 6 years old are not asked the question on difficulties learning or understanding. We do not consider that question as missing.

S1 Table in the supporting information provides GDP per capita and age composition for the eight countries in our sample. For example, in 2010, the Dominican Republic had a GDP per capita of 5600 dollars, and only 12 percent of the population was over the age of 55. Uruguay had about twice as large a GDP per capita and a share of the population over 55 as the Dominican Republic.

IPUMS provides systematic random samples drawn from all households that participated in the census. Table 1 describes the main features of the samples. Sample sizes vary from 116,917 in Trinidad and Tobago (2011), representing 8.8 percent of the total population, to 11,938,402 in Mexico (2010), representing 10 percent of the population. All samples were drawn with household clustering and include information on all individuals in the selected households. S2 Table in the supporting information describes the main features of the sampling methods used to draw each sample. To compute our estimates, we use data on disability, education, and employment status, which we define below.

**Disability.** In recent decades, the notion of disability has evolved from a strict medical concept to the social model which incorporates the interaction between a person's impairment and the barriers to functioning in his or her environment [11]. According to this approach embodied in the CRPD and the WHO's International Classification of Functioning, Disability, and Health, a person's disability is defined as a dynamic interaction between an impairment and external factors; on its own, a health condition or impairment is not a disability [12]. In line with this definition, the Washington Group has piloted, validated, and endorsed a set of questions for measuring disability in national censuses and household surveys. The questions inquire about difficulties in performing basic activities such as walking, seeing, hearing, cognition, communication, and self-care.

For our analysis, we construct a disability variable that maximizes comparability across countries, restricting to the WG questions when available. Our disability variable takes the value of one if the respondent answered affirmatively to any disability question included in the census and zero otherwise. We code as missing all respondents for whom, based on the available information, we cannot establish their disability status. Respondents who responded positively to a question are coded as having a disability, even if they did not respond to all questions. Respondents who did not respond to any of the disability questions or who

responded negatively to some but left others unanswered are coded as missing. Table 1 (last column) reports the proportion of individuals we code as missing. In Uruguay 2011, the question of "difficulties understanding or learning" is not asked to children under six years old. We do not consider that question as missing. In this case, we assess the disability status of these children by checking the other three questions in the survey. Table S3 Table presents the characteristics of individuals who were assigned a disability status (non-missing) with those considered "missing" for the Dominican Republic, Trinidad and Tobago, and Uruguay, where missing values exceed 1 percent of the observations. We do not observe major differences in age and sex composition between "missing" and "non-missing" observations in any of the countries.

The survey questions in Brazil (2010), Trinidad and Tobago (2011), and Uruguay (2011) inquire about the degree of the difficulty or limitation. In these countries, we consider indicating "some difficulty" or more as an affirmative answer. The response categories are: "no difficulty", "some difficulty", "a lot of difficulty", and "cannot do it at all". We chose this operationalization of disability because of theoretical and practical reasons. First, it follows the expert Washington Group's advice regarding the most appropriate threshold to define individuals with and without a disability [13]. Second, it increases the comparability with other countries in our sample, including those that apply the WG questions without gradation of difficulty. For example, we consider that the natural comparison of the binary responses to the question in Mexico (2010) "Does [the respondent] have difficulty . . ." is the threshold of "some difficulty" rather than "a lot of difficulty". Finally, it allows for comparisons with other countries such as Canada, which reports disability prevalence using the same threshold of "some difficulty", and the United States that does not inquire about gradations of difficulties and follows the thresholds used in other publications [14]. The U.S. Census Bureau guides disability data users in the website: https://www.census.gov/topics/health/disability/guidance.html (accessed November 9, 2020). Several countries have included or will include questions on the severity of the limitations in the 2020 census round. Once these data become available, it will be able to compare disability prevalence using the higher thresholds. Our disability indicator differs slightly from the harmonized variable provided by [10]. We constructed our disability indicator to increase comparability across countries and adhere more closely to the WG recommendations on disability characterization. The differences between our variable and the harmonized variable are due to three reasons: First, in Brazil and Uruguay (but not in Trinidad and Tobago), the harmonized variable does not consider "some difficulty" as a disability. We consider "some difficulty" as a disability in all three countries. Second, some countries include a filter question (e.g., Ecuador) or a general question (e.g., Panama) on disability status. The harmonized IPUMS variable takes the responses to this question to define disability. We construct our variable using responses to questions asking about difficulties or limitations in performing certain activities or tasks. Finally, we code respondents who responded positively to at least one of these questions as having a disability, even if they did not respond to all questions. The harmonized variable codes respondents who did not respond to a question as "unknown". Nevertheless, for completeness and comparability, we also present estimates for the overall disability prevalence rates considering only individuals who reported "a lot of difficulty" or more as an affirmative answer.

We also construct a set of indicator variables that consider separately different types of difficulties: vision, hearing, motor, and cognitive. In the case of vision and hearing disabilities, all surveys include specific questions or categories referring to them. For motor disability, we consider only those questions that refer to difficulties walking or moving upper or lower limbs. Finally, cognitive disability is defined using questions and categories related to difficulties with concentrating, learning, understanding, or remembering, and those directly referring to

specific diagnoses such as Down Syndrome. We exclude limitations associated with psycho-social conditions unless they are included in the same category as other cognitive limitations (such as in Brazil (2010) and Trinidad and Tobago (2011)). The WG short list of questions does not fully capture psycho-social disabilities. Research in the U.S. has found that the short list fails to capture approximately half of people with psycho-social disabilities. An extended list of questions has been developed to improve the coverage of this under-measured population [15, 16].

Seven of the eight census questionnaires applied the "functional approach" to measuring disability, only Ecuador applied a fully medical approach. However, none of the seven countries strictly applied the Washington Group questions. Only Uruguay and Brazil inquired about gradations of disability rather than a binary, yes or no questions. Research suggests that when a binary scale is used, less severe cases of disability are not reported [17]. In all cases, modifications were made to WG questions and additional questions or filters may have influenced measurement. The questionnaires in Ecuador, Panama, and Trinidad and Tobago explicitly use the word "disabilities". As suggested by [6, 15], this latter approach might induce lower response rates, especially in societies where a stigma applies to the word disability. More-over, the word "disability" may be understood differently across individuals and cultures. For example, older adults with significant limitations in doing certain basic activities may not con-sider themselves as having a disability if they can function as expected for their age [6]. In the Trinidad and Tobago questionnaire, a filter-type question about whether the respondent suf-fers from a long-standing disability may have suppressed responses to the subsequent WG questions.

The approach used to ask about disabilities is not the only difference between surveys: the range of activities covered by the survey and the precise wording of the questions vary across countries sharing the same approach. While the WG questions intentionally avoid the use of words that may be considered labels, we note that all seven countries that implemented the four basic WG questions also mixed in a medical approach to a varied extent through the wording of instructions or additional questions about impairments or diagnoses. For example, the Costa Rica (2011) survey refers to diagnoses, specifically Down Syndrome, bipolar, and schizophrenia. The variations in the instruments (together with cultural differences that can affect the interpretation of the questions) are essential to assess the differences in the preva-lence of disability between countries. S4 Table in the supporting information provides details about the wording of the questions to measure disability in each census and provides the cur-rent WG's Short Set on Functioning (WG-SS) as a reference. We note that some of the termi-nologies in these questions may be considered derogatory but have chosen to maintain the translation provided by IPUMs. This translation is accurate and captures that, in some cases, the original Spanish version included terms that might be considered derogatory and might have affected reporting.

**Education.** All samples include harmonized questions on school attendance and educa-tional attainment. We use this information to estimate attendance and completion rates for different subpopulations and assess the gap in access to education between people with and without a disability. Information on educational attainment refers to the highest completed level of schooling: less than primary completed, primary completed, secondary completed, or university completed.

We also use this information to construct a variable indicating the educational level of the household head. We use this variable to approximate the socioeconomic situation of the household, in the absence of comparable and reliable information on income, assets or expenditures.

**Employment status.** We use the information on the employment situation to calculate employment rates among people with and without disabilities. The information on the labor situation comes from the harmonized variables provided by IPUMS International, based on the censuses' questions. The information indicates whether the respondent was working in a specific period (in most cases, the week before the census).

**Other information.** We use information on respondents' age (in years) and sex.

## Estimation method

**Disability prevalence.** We estimate the prevalence of disability in each country using representative samples from the population census. Our estimates account for the sampling method used to obtain each sample (i.e., clustering, stratification, and design weights). The prevalence rate is simply the ratio of the total number of individuals of a population with a disability over the total number of individuals in the population. We provide estimates of prevalence for the whole population (ages 3 and older), specific groups of the population, and different types of disabilities. We estimate each population total with the [18] estimator, using the sampling weights provided by IPUMS-International for each person in the sample. We compute confidence intervals using the [19] method with the adjustments proposed by [20] and [21].

When computing prevalence estimators for some subpopulations, sampling variability can be large, and point estimates and their variances might be imprecisely estimated. We follow the presentation guidelines proposed by the National Center for Health Statistics (NCHS) Data Suppression Workgroup [22]. Unlike smaller sampling frames such as the Demographic Health Surveys or specific disability surveys, the use of census samples allows us to characterize small age groups.

**Gaps between sexes.** To compute sex gaps in disability prevalence, we estimate linear regression models for each country using a disability indicator as the dependent variable and a female indicator variable as a independent variable. We include cohort fixed effects to control for potential differences in age structure between sexes. Formally, the model we estimate is:

$$y_{ci} = \alpha_c + \beta F_{ci} + \epsilon_{ci} \tag{1}$$

where $y_{ci}$ is the value for the outcome of interest (e.g. disability status) for individual $i$ in cohort $c$; $F_{ci}$ is an indicator variable taking value one for female respondents; $\alpha_c$ is the cohort fixed effect; and $\epsilon_{ci}$ is an idiosyncratic error term. The cohort is given by the respondent's age at the moment of the survey. The coefficient of interest is $\beta$, which captures the average sex gap in the dependent variable, after accounting for systematic differences across cohorts. This coefficient is (approximately) the population-weighted average of the sex gap in the disability prevalence rates across cohorts.

**Disability gaps in education and employment.** We use different measures of educational attainment and employment status to compute disability gaps in education and labor market opportunities. For educational outcomes, we use binary indicators on: (a) school attendance among children in primary school age (6 to 11 years old); (b) school attendance among people in secondary school age (12 to 17 years old); and (c) secondary school completion among people between 25 and 34 years old. For labor outcomes, we focus on whether the person is employed or not (binary outcome) and examine separately two age groups: young adults between 25 and 34 years old and middle-aged adults from 35 to 54.

For each of these measures, we report three statistics. The first two are the average outcome for individuals living with and without a disability, respectively. Since our measures are all binary (taking values 0 or 1), we compute averages by taking the ratio of the total number of

individuals that responded affirmatively (e.g. indicate that are attending school) over the total number of individuals in the population. The third statistic is an estimate of the disability gap, which we obtain from the following linear regression model:

$$y_i = \alpha + \gamma I_i + \epsilon_i \tag{2}$$

where $y_i$ is the value for the outcome of interest for individual $i$; $I_i$ is an indicator variable denoting if the respondent has a disability; $\alpha$ is the regression intercept; $\epsilon_i$ is an idiosyncratic error term. The coefficient of interest is $\gamma$, which captures the average disability gap in the dependent variable. In these analyses, we focus on relatively narrow age groups and do not include cohortage fixed effects.

**Estimation.** We estimated all our point estimates and standard errors using Stata, version 15.1. We follow IPUMS guidelines and use Stata's survey estimation commands (`svyset` and `svy`), defining variable "perwt" as sampling weights, variable "serial" as cluster identifier, and variable "strata" to define (pseudo) strata. IPUMS guidelines are available at: https://international.ipums.org/international/variance_estimation.shtml (accessed October 20th, 2020). The technical appendix (S1 Appendix) provides more details about the estimation of prevalence rates, computation of confidence intervals, and suppression rules.

**Extrapolations and projections.** In our discussion section, we use the estimated prevalence rates in Uruguay (2011) (and the Dominican Republic (2010) in the Supporting information) to make linear extrapolations of the overall disability prevalence to other years and other countries in LAC based on their demographic structure. Section 5 ("Estimating Total Population with Disability: Extrapolation") in the technical appendix (S1 Appendix) provides more details about this exercise.

## Results

### Prevalence of disability: Estimates by country

The prevalence of disability in the population (ages 3 and older) varies greatly between countries, ranging from 4.5 percent in Trinidad and Tobago (2011) to 24.9 percent in Brazil (2010). Fig 1 shows the estimated prevalence rates for each country among individuals aged 3 and older. S5 Table report overall prevalence rates for individuals aged 15 and older. In recent years, the WG, together with UNICEF, has generated new sets of questions to measure functioning among children ages 2 to 4 and 5 to 17. These modules are intended to provide more precise estimates of functional difficulties among children, for whom the standard WG questions might be less reliable. This module has not yet been widely implemented in LAC and, therefore, could not be used in our study. Table 2 reports these prevalence rates and their 95-percent confidence intervals (columns 1 to 3). In columns 4 to 9 of Table 2, we provide estimates by sex. In six out of eight countries, the prevalence of disability is higher for women than for men. Some of the differences we observe across countries are expected given the approach used to define disability and the instructions' wording. Countries reporting the lowest disability prevalence are those in which questions refer explicitly to the presence of specific medical conditions and diverge widely from the "functional approach" (Ecuador (2010)) or refer to disabilities in a filter-type question preceding the WG questions (Trinidad and Tobago (2011)). It is also possible that appending questions to the recommended Washington Group short list or including diagnoses can prompt stigma. In Mexico's (2010) and Panama's (2010) cases, the surveys included additional questions about types of disability and causes of impairment, which could have reduced disclosure. S6 Table presents the estimates of the overall disability prevalence rates in Brazil (2010), Trinidad and Tobago (2011) and Uruguay (2011) considering only individuals who reported "a lot of difficulty" or more as an affirmative

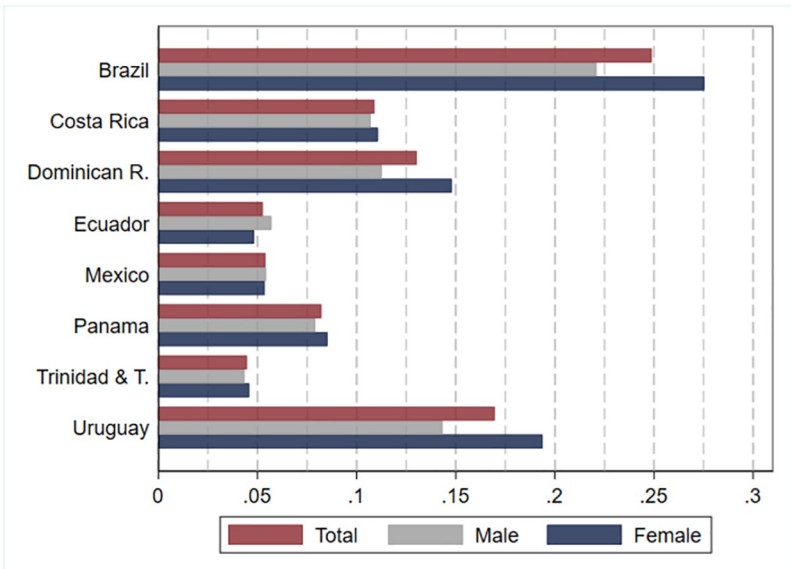

**Fig 1. Prevalence of disability: Estimates by country and sex.** Source: authors' estimations based on data provided by [10] from censuses collected by National Statistics Offices in each country. Estimates for Brazil, Dominican Republic, Ecuador, Mexico, and Panama refer to the year 2010. Estimates for Costa Rica, Trinidad and Tobago, and Uruguay refer to the year 2011. Estimates consider individuals aged 3 and older.

answer. Using this definition of disability, prevalence rates fall to 6.9 percent in Brazil (2010), 5 percent in Uruguay (2011) and 2.4 percent in Trinidad and Tobago (2011).

Fig 2 shows the prevalence of disability for different age groups (5-year intervals until 80 years old and a group for people 81 years old or older) and sex in each country. The prevalence of disability increases with age at an accelerating rate in our eight countries. As shown, children between 3 and 5 years of age have a disability rate of 0.79 percent in Trinidad and Tobago (2011) and 3.8 percent in Brazil (2010), while in the oldest age group (56+ years), the range goes from 12.4 percent to 60.4 percent respectively. In most of the following analysis and in Discussion section, we assume that differences respond to age. However, it is plausible that cohort effects explain at least part of the variation.

**Table 2. Prevalence of disability by country and gender: Estimates.**

| | Both Sexes | | Men | | Women | |
|---|---|---|---|---|---|---|
| | Est. | 95% C.I. | Est. | 95% C.I. | Est. | 95% C.I. |
| Brazil | 24.9 | [24.8 24.9] | 22.1 | [22.0 22.1] | 27.5 | [27.5 27.6] |
| Costa Rica | 10.9 | [10.8 11.0] | 10.7 | [10.6 10.9] | 11.1 | [10.9 11.2] |
| Dominican R. | 13.0 | [12.9 13.1] | 11.3 | [11.2 11.4] | 14.8 | [14.7 14.9] |
| Ecuador | 5.26 | [5.22 5.30] | 5.70 | [5.64 5.76] | 4.83 | [4.78 4.88] |
| Mexico | 5.39 | [5.35 5.44] | 5.43 | [5.35 5.51] | 5.36 | [5.32 5.41] |
| Panama | 8.22 | [8.11 8.33] | 7.92 | [7.78 8.05] | 8.53 | [8.38 8.67] |
| Trinidad & T. | 4.46 | [4.32 4.60] | 4.34 | [4.16 4.53] | 4.58 | [4.39 4.77] |
| Uruguay | 17.0 | [16.8 17.1] | 14.3 | [14.1 14.5] | 19.4 | [19.2 19.6] |

Source: authors' estimations based on data provided by [10] from censuses collected by National Statistics Offices in each country. Estimates for Brazil, Dominican Republic, Ecuador, Mexico, and Panama refer to the year 2010. Estimates for Costa Rica, Trinidad and Tobago, and Uruguay refer to the year 2011. Estimates consider individuals aged 3 years and older.

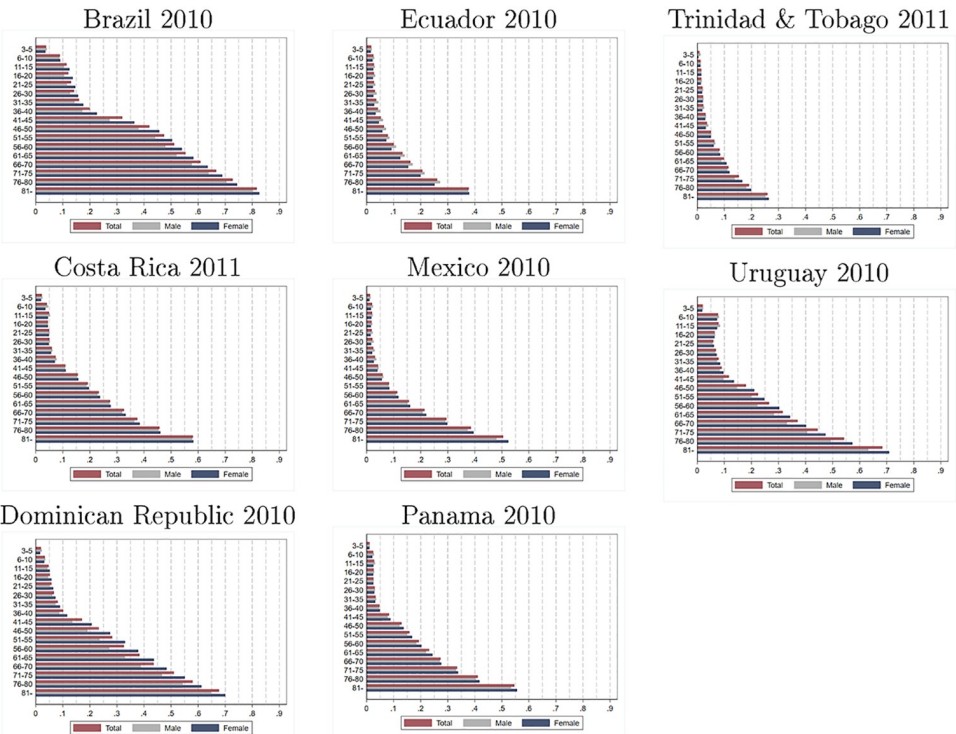

**Fig 2. Prevalence of disability: Estimates by age group and sex, by country.** Source: authors' estimations based on data provided by [10] from censuses collected by National Statistics Offices in each country. Estimates for Brazil, Dominican Republic, Ecuador, Mexico, and Panama refer to the year 2010. Estimates for Costa Rica, Trinidad and Tobago, and Uruguay refer to the year 2011. Estimates consider individuals aged 3 and older.

An interesting pattern which we can also observe in Fig 2 is that the relative disability rate by sex varies with age in many countries. For example, in Mexico, the disability prevalence among the 3 to 5 year-olds is 38 percent larger for boys. This gap falls to zero in the age bracket 51 to 55. Among those older than 81, the disability rate for women is 9 percent larger than for men. To parsimoniously model these age and sex patterns, we estimate Eq (1) for each country and different age groups. Table 3 presents the results of these estimations.

In the young population (under 30 years old, first column), the prevalence is generally higher among men than among women. Estimates show a negative (that is, higher among men) and statistically significant gap (at standard significance levels) in five of the eight countries. In two of the remaining countries (Brazil and the Dominican Republic), the gap is positive. For the older population (over 64 years old, third column), the patterns reverse: the prevalence is generally higher among women than among men. Estimates show a positive and statistically significant gap (at standard significance levels) in six of the eight countries studied.

Only in Brazil and the Dominican Republic females have a higher prevalence of disability at all ages. In the remaining countries, the sex patterns vary by age. Therefore, not surprisingly, the results are inconclusive when considering all age groups together (column four). In four countries (Brazil, Dominican Republic, Panama, and Uruguay), the prevalence of disability is significantly higher among women than men; in two countries (Ecuador and Mexico), the opposite occurs: the gap is positive for men. In Costa Rica and Trinidad and Tobago, the difference is not statistically different from zero.

Our previous analysis aggregated all types of disabilities. Different types of disabilities sometimes require different policy actions, so it is important to consider them separately. We

**Table 3. Gender differences in disability prevalence by age group: Estimates.**

| | 3–30 | | 30–64 | | $\geq$65 | | $\geq$3 | |
|---|---|---|---|---|---|---|---|---|
| | Preval. Rate | $\beta$ | Preval. Rate | $\beta$ | Preval. Rate | $\beta$ | Preval. Rate | $\beta$ |
| Brazil | 11.2 | 0.021[‡] | 33.6 | 0.062[‡] | 67.7 | 0.047[‡] | 24.9 | 0.040[‡] |
| | | (0.00035) | | (0.00047) | | (0.0012) | | (0.00027) |
| Costa Rica | 4.45 | -0.0049[‡] | 13.5 | 0.0017 | 40.9 | 0.010[*] | 10.9 | -0.0011 |
| | | (0.00089) | | (0.0015) | | (0.0051) | | (0.00085) |
| Dominican R. | 4.74 | 0.0064[‡] | 18.8 | 0.062[‡] | 52.3 | 0.079[‡] | 13.0 | 0.031[‡] |
| | | (0.00060) | | (0.0012) | | (0.0039) | | (0.00061) |
| Ecuador | 2.63 | -0.0074[‡] | 6.15 | -0.015[‡] | 22.8 | -0.013[‡] | 5.26 | -0.011[‡] |
| | | (0.00036) | | (0.00067) | | (0.0026) | | (0.00036) |
| Mexico | 1.89 | -0.0064[‡] | 5.71 | -0.0035[‡] | 31.2 | 0.016[‡] | 5.39 | -0.0037[‡] |
| | | (0.00024) | | (0.00051) | | (0.0041) | | (0.00036) |
| Panama | 2.50 | -0.0036[‡] | 10.3 | 0.010[‡] | 36.4 | 0.0080 | 8.22 | 0.0027[‡] |
| | | (0.00076) | | (0.0016) | | (0.0056) | | (0.00084) |
| Trinidad & To. | 1.61 | -0.0026[†] | 5.00 | -0.0017 | 15.7 | 0.013[*] | 4.46 | -0.00077 |
| | | (0.0012) | | (0.0019) | | (0.0070) | | (0.0012) |
| Uruguay | 6.47 | -0.0021 | 16.6 | 0.042[‡] | 49.0 | 0.070[‡] | 17.0 | 0.027[‡] |
| | | (0.0013) | | (0.0019) | | (0.0044) | | (0.0011) |

Source: authors' estimations based on data provided by [10] from censuses collected by National Statistics Offices in each country. Estimates for Brazil, Dominican Republic, Ecuador, Mexico, and Panama refer to the year 2010. Estimates for Costa Rica, Trinidad and Tobago, and Uruguay refer to the year 2011. The table reports the difference in disability prevalence prevalence across genders, accounting for the age of the respondents. We estimate a linear regression model using our disability indicator as dependent variable, and a female indicator variable and cohort fixed effects as independent variables. Reported $\beta$ are the OLS estimates of the coefficient for the female indicator variable. Each figure in the table comes from a different regression. "Prevalence rates" figure are the estimated overall disability prevalence rates for individuals in the age group. Each pair of columns reports estimates for a different age group. Symbols denote statistical significance:

[*] 0.10,

[†] 0.05, and

[‡] 0.01.

study the prevalence in the population of four different types of limitations: vision, hearing, motor, and cognitive. Fig 3 shows disability prevalence rates for each of these four categories by country and by sex. S7 and S8 Tables in the supporting information report estimates and their 95-percent confidence intervals for the full population and four different age groups (3 to 5, 6 to 17, 18 to 55, and older than 56 years).

Our estimations indicate that vision and motor disabilities have the highest prevalence in the countries examined. The prevalence of vision disability ranks first in 6 out of 8 countries in the entire population, ranging from 1.4 percent in Ecuador to 19.6 percent in Brazil. Motor limitations rank first in the remaining two countries (Ecuador and Mexico) and range from 2.1 percent (Trinidad and Tobago) to 7.2 percent (Brazil). For cognitive limitations, prevalence rates range from 0.2 percent in Mexico to 3 percent in the Dominican Republic. The prevalence of hearing limitations ranges from 0.5 percent (Trinidad and Tobago) to 5.3 percent (Brazil).

Estimates of the prevalence of different limitations by age groups reveal somewhat different patterns. Motor, visual, and hearing disabilities have a strong and clear age gradient. The prevalence rate of motor impairment among people over 55 years old is, on average, 7.2 higher than in people between 18 and 55 years old. This ratio is, on average, 4.5 for visual impairment and 8.3 for hearing impairment. The age gradient is positive but weaker for intellectual

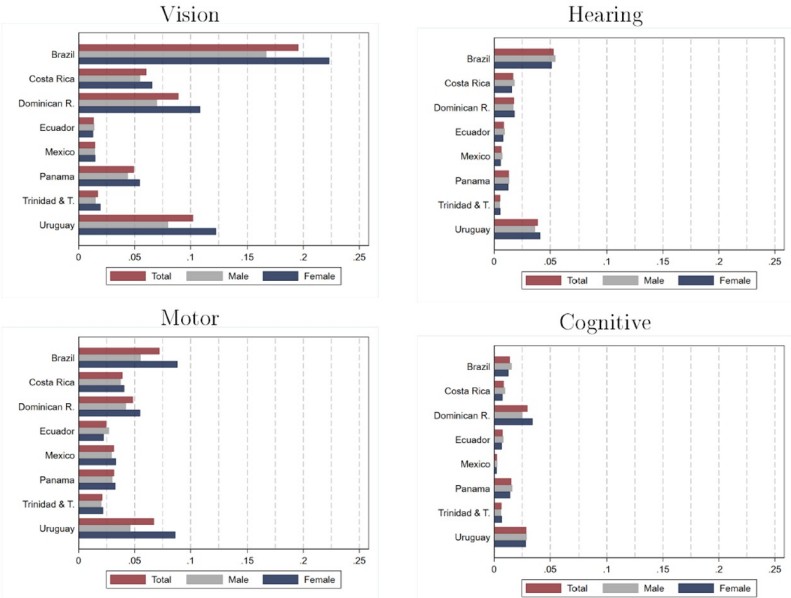

**Fig 3. Prevalence of disability: Estimates by type, country and sex.** Source: authors' estimations based on data provided by [10] from censuses collected by National Statistics Offices in each country. Estimates for Brazil, Dominican Republic, Ecuador, Mexico, and Panama refer to the year 2010. Estimates for Costa Rica, Trinidad and Tobago, and Uruguay refer to the year 2011. Estimates consider individuals aged 3 years and older.

limitations: the prevalence among people over 55 is "only" 2.8 times that among people between 18 and 55.

Finally, we study the socioeconomic gradient in the prevalence of disability by separating households according to the head's highest level of education. This variable usually correlates with other relevant socioeconomic indicators such as family income, the level of education attained by other household members, and access to services. Following IPUMS classification, we consider four education levels: primary incomplete, primary complete, secondary complete, and college complete. Fig 4 shows the prevalence of disability for people in these groups of households in each of the countries in our study. S9 Table in the supporting information shows the estimates and their 95-percent confidence interval. It also reports the estimates for men and women separately.

There is a strong socioeconomic gradient of disability in all countries. The people with the highest prevalence of disability live in households whose head has not completed primary school. The prevalence of disability is significantly lower in households whose head completed primary school and even more so in households whose head completed secondary school. For example, individuals in households where the head has not completed primary education are, on average, more than two times more likely to experience disability than individuals in households where the head has completed a college degree.

The strong socioeconomic gradient in the prevalence of disability persists when considering specific age groups and different types of disabilities. Fig 5 (and S9 Table) show the estimates of the prevalence of disability for persons from 3 to 5 years old, from 6 to 17 years old, from 18 to 55 years old, and over 55 years old for the different groups of households considered previously. In all age groups for practically all countries, the prevalence of disability is significantly higher in households with low education levels.

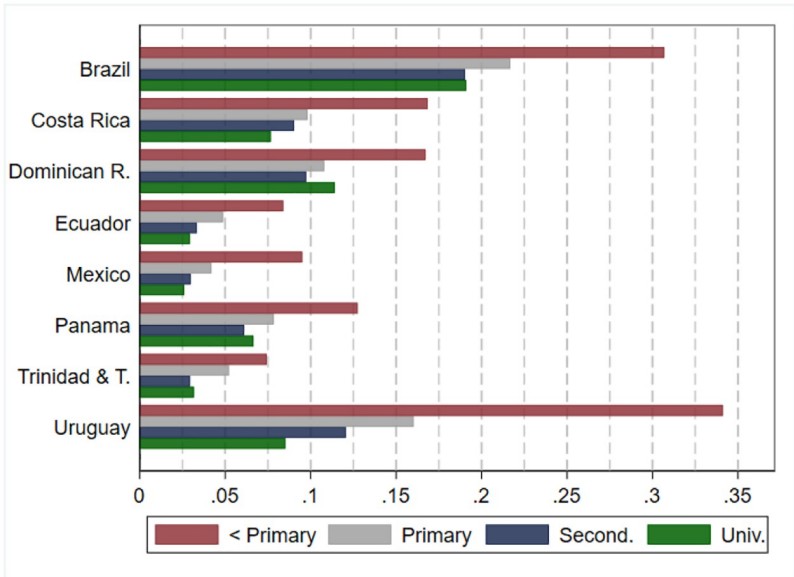

**Fig 4. Prevalence of disability: Estimates by country and household head education level.** Source: authors' estimations based on data provided by [10] from censuses collected by National Statistics Offices in each country. Estimates for Brazil, Dominican Republic, Ecuador, Mexico, and Panama refer to the year 2010. Estimates for Costa Rica, Trinidad and Tobago, and Uruguay refer to the year 2011. Estimates consider individuals aged 3 years and older.

Fig 6 shows the results distinguishing by type of disability (grouping all ages together). The graph confirms that for all types of disability considered (visual, motor, hearing, and cognitive), the prevalence is significantly higher in those households where the head of the household has a low level of education.

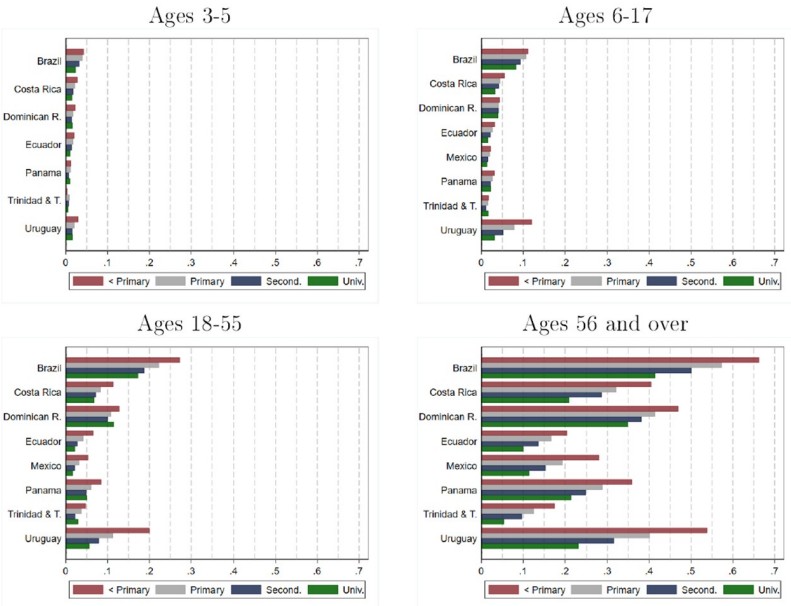

**Fig 5. Prevalence of disability: Estimates by age, country and household head education level.** Source: authors' estimations based on data provided by [10] from censuses collected by National Statistics Offices in each country. Estimates for Brazil, Dominican Republic, Ecuador, Mexico, and Panama refer to the year 2010. Estimates for Costa Rica, Trinidad and Tobago, and Uruguay refer to the year 2011.

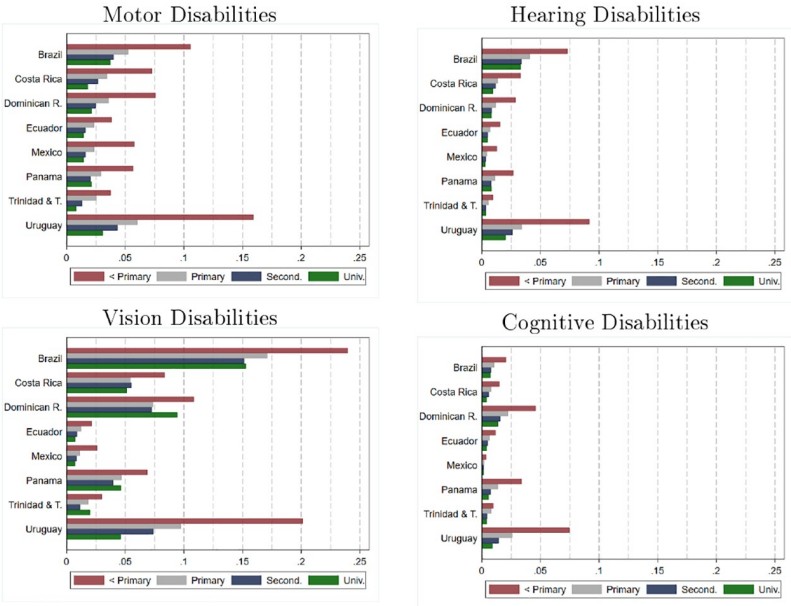

**Fig 6. Prevalence of disability: Estimates by type, country and household head education level.** Source: authors' estimations based on data provided by [10] from censuses collected by National Statistics Offices in each country. Estimates for Brazil, Dominican Republic, Ecuador, Mexico, and Panama refer to the year 2010. Estimates for Costa Rica, Trinidad and Tobago, and Uruguay refer to the year 2011. Estimates consider individuals aged 3 years and older.

## Disability gaps in education and employment

The design of policies to achieve the tenth Sustainable Development Goal's target of reducing the number of people with disabilities living below 50 percent of median income requires information on this population's educational and employment opportunities. As indicated above, we assess the disability gap in educational outcomes using three measures: (a) school attendance among children in primary school age (6 to 11 years old); (b) school attendance among people in secondary school age (12 to 17 years old); and (c) secondary school completion among people between 25 and 34 years old.

In Table 4, we report the average outcome for individuals living with and without a disability, respectively, and the disability gap estimated using Eq (2).

School attendance among children who report no disabilities exceeds 94 percent in all countries, but the situation is significantly different for children with disabilities. There is a systematic gap in school attendance among primary school-age children: in all countries except for Costa Rica, school attendance is significantly lower among children with disabilities. The average gap is eight percentage points and ranges from -1.2 percentage points in Costa Rica to 17.0 percent in Trinidad and Tobago.

We also document a significant disability gap in school attendance for secondary school-age people (12 to 17 years old). Columns four to six in Table 4 show the estimates for school attendance rates and gaps in this age group: the average school attendance rate is 85.9 percent for adolescents without disabilities and 75.7 percent for adolescents with disabilities. The disability gap averages 10.3 percentage points and ranges from 2.6 percentage points in Costa Rica to 19.7 percentage points in Trinidad and Tobago.

We calculate secondary school completion rates and the disability gap for people between 25 and 34. Columns 7 to 9 in Table 4 show the results for the different countries. The estimates reveal a significant disability gap in secondary school completion in all the countries included

**Table 4. Estimates for school attendance and completion rates and gaps.**

| | Attendance: 6–11 | | | Attendance: 12–17 | | | Completion: 25–34 | | |
|---|---|---|---|---|---|---|---|---|---|
| | **(1) No disab.** | **(2) Any disab.** | **(3) Gap** | **(4) No disab.** | **(5) Any disab.** | **(6) Gap** | **(7) No disab.** | **(8) Any disab.** | **(9) Gap** |
| Brazil | 0.973 | 0.957 | 0.016[‡] | 0.897 | 0.883 | 0.015[‡] | 0.505 | 0.440 | 0.065[‡] |
| | | | (0.001) | | | (0.001) | | | (0.001) |
| Costa Rica | 0.942 | 0.955 | -0.012[†] | 0.838 | 0.812 | 0.026[‡] | 0.436 | 0.383 | 0.054[‡] |
| | | | (0.005) | | | (0.009) | | | (0.008) |
| Dominican R. | 0.956 | 0.828 | 0.129[‡] | 0.873 | 0.762 | 0.111[‡] | 0.442 | 0.385 | 0.057[‡] |
| | | | (0.006) | | | (0.006) | | | (0.005) |
| Ecuador | 0.970 | 0.826 | 0.144[‡] | 0.834 | 0.652 | 0.182[‡] | 0.487 | 0.281 | 0.206[‡] |
| | | | (0.006) | | | (0.007) | | | (0.006) |
| Mexico | 0.970 | 0.827 | 0.143[‡] | 0.794 | 0.618 | 0.176[‡] | 0.386 | 0.188 | 0.197[‡] |
| | | | (0.005) | | | (0.006) | | | (0.006) |
| Panama | 0.974 | 0.942 | 0.032[‡] | 0.883 | 0.833 | 0.049[‡] | 0.538 | 0.432 | 0.106[‡] |
| | | | (0.007) | | | (0.012) | | | (0.013) |
| Trinidad & T | 0.989 | 0.819 | 0.170[‡] | 0.914 | 0.717 | 0.197[‡] | 0.787 | 0.539 | 0.248[‡] |
| | | | (0.038) | | | (0.040) | | | (0.026) |
| Uruguay | 0.992 | 0.976 | 0.016[‡] | 0.841 | 0.775 | 0.065[‡] | 0.392 | 0.259 | 0.133[‡] |
| | | | (0.003) | | | (0.009) | | | (0.008) |
| Average | 0.971 | 0.891 | 0.080 | 0.859 | 0.757 | 0.103 | 0.497 | 0.363 | 0.133 |

Source: authors' estimations based on data provided by [10] from censuses collected by National Statistics Offices in each country. Estimates for Brazil, Dominican Republic, Ecuador, Mexico, and Panama refer to year 2010. Estimates for Costa Rica, Trinidad and Tobago, and Uruguay refer to year 2011. Symbols in columns (3), (6) and (9) denote statistical significance:

[*] 0.10,

[†] 0.05, and

[‡] 0.01.

in our study. Among people with disabilities, the secondary completion rate is, on average, only 36.3 percent. Mexico shows the lowest completion rate among persons with disabilities (18.8 percent) and Trinidad and Tobago, the highest (53.9 percent). The average disability gap in secondary school completion is 13.3 percentage points.

S10 Table presents the disability gaps in school attendance and completion by sex. The disability gap in school attendance is similar for primary school-age boys and girls: the average disability gap is 7.2 percent for girls and 8.7 percent for boys. For those of secondary school age, the disability gap in school attendance is generally larger for boys (12.1 percent on average) than for girls (8.4 percent on average). The disability gap in school completion rates for 25–34 year olds is similar between the two sexes (13.2 percent for females and 13.5 for males, on average).

S11 Table shows the disability gaps in school attendance and completion by type of disability. The gap in school attendance varies significantly between different types of disabilities. The average gap is the largest for children with cognitive difficulties (13 percent among primary school-age children and 19.9 percent among secondary school-age ones) and motor limitations (15.5 percent and 16.9 percent), followed by those with hearing limitations (8.3 percent and 10.1 percent). In the case of children with vision disabilities, we observe a positive but smaller average school attendance gap among primary school age children (2.6 percent) and no systematic gap among secondary school age children. The patterns are similar, but more pronounced, when looking at school completion gaps among individuals aged 25 to 34.

Disability gaps in school attendance and completion are large and significant for both sexes in almost every country in our study. It is important to note that this analysis has considered inclusion at the extensive margin, i.e., whether children with disabilities are being excluded from educational facilities. The data available in the census is not sufficient to assess whether students are receiving inclusive education in mainstream schools. The results suggest a continued need to reduce the gap in access to education. The more positive note is that some countries such as Brazil, Costa Rica and Uruguay have closed the gap in attendance at primary and some countries have made important strides at secondary, although not at completion. The CRPD does not contemplate that severity of difficulty should be correlated with inclusion. In practice however, the estimated gaps may be reflecting differential access based on gradation of disability. Differences in measurement can also affect results across countries. If some of this relative success is due to their inclusion policies, the lower-performing countries can significantly improve by adopting and adapting them.

Finally, we examine the employment situation of people with disabilities in the eight countries included in our study and compare it with the employment situation of those who do not report a disability. We focus on two age groups: young adults between 25 and 34 years old and middle-aged adults from 35 to 54. In this analysis, we use the information on the employment status provided by IPUMS International, which classifies people as employed, unemployed (not working but searching for a job), and inactive (not working and not looking for work). We estimate the percentage of employed people with and without disabilities and, as for educational outcomes, calculate the difference as a measure of the disability gap in the access to labor market opportunities.

The first three columns in Table 5 report employment rates (that is, the ratio between persons employed over total population -active and inactive-) for persons aged 25–34 with and without disabilities and the difference between them. The estimates show that the employment-to-population ratio is significantly lower among persons with disabilities in all the countries included in our study. The employment rate for persons without disabilities aged 25–34 is, on average, 70.4 percent. The country with the lowest rate is the Dominican Republic, where 58.5 percent of the people with no disabilities are employed. Uruguay shows the highest rate, 82.9 percent, followed by Trinidad and Tobago (77.8 percent). Among persons with disabilities in the same age range, the employment rate is, on average, only 51.9 percent, ranging from 37.1 percent in Trinidad and Tobago to 63.8 percent in Brazil.

The employment disability gap for people aged 25–34 is, on average, 18.5 percentage points. The gap is greater than ten percentage points in almost every country included in the study (with the exceptions of the Dominican Republic and Brazil), reflecting the significant barriers to employment in the region for persons with disabilities. Trinidad and Tobago (40.7 percentage points) and Mexico (23) show the largest disability gaps in employment. The analysis for the population between 35 and 54 years old shows very similar results. Columns 4 to 6 in Table 5 show the employment rates and the disability gap in employment for this age group. The average disability gap in the employment rate is 18.6 percentage points.

S12 Table presents the employment rates and disability gaps by sex. Employment rates are generally much lower for women than men, both with and without disabilities. The disability gap is also lower among women than men, but still large and significant for both sexes in all countries.

S13 Table presents the employment rates and disability gaps by type of disability. We observe large and significant gaps for the four types of disabilities we examine: vision, hearing, motor and cognitive. Similarly to the patterns observed in access to education, the average employment gap in the countries in our sample is largest for individuals with cognitive difficulties (45.1 percent for persons aged 25–34 and 45.9 for persons aged 35–54), followed by

**Table 5. Estimates for employment rates and disability gaps.**

| | Employment Rate: 25–34 | | | Employment Rate: 35–54 | | |
|---|---|---|---|---|---|---|
| | **(1) No disab.** | **(2) Any disab.** | **(3) Gap** | **(4) No disab.** | **(5) Any disab.** | **(6) Gap** |
| Brazil | 0.724 | 0.638 | 0.086[‡] | 0.740 | 0.620 | 0.119[‡] |
| | | | (0.001) | | | (0.001) |
| Costa Rica | 0.639 | 0.520 | 0.119[‡] | 0.624 | 0.480 | 0.144[‡] |
| | | | (0.008) | | | (0.005) |
| Dominican R. | 0.585 | 0.495 | 0.090[‡] | 0.589 | 0.473 | 0.116[‡] |
| | | | (0.005) | | | (0.003) |
| Ecuador | 0.698 | 0.509 | 0.188[‡] | 0.718 | 0.523 | 0.196[‡] |
| | | | (0.006) | | | (0.004) |
| Mexico | 0.667 | 0.437 | 0.230[‡] | 0.680 | 0.488 | 0.192[‡] |
| | | | (0.006) | | | (0.003) |
| Panama | 0.694 | 0.551 | 0.142[‡] | 0.720 | 0.591 | 0.129[‡] |
| | | | (0.013) | | | (0.006) |
| Trinidad & T | 0.778 | 0.371 | 0.407[‡] | 0.778 | 0.347 | 0.431[‡] |
| | | | (0.025) | | | (0.014) |
| Uruguay | 0.829 | 0.648 | 0.181[‡] | 0.848 | 0.666 | 0.182[‡] |
| | | | (0.009) | | | (0.005) |
| Average | 0.702 | 0.521 | 0.180 | 0.712 | 0.523 | 0.189 |

Source: authors' estimations based on data provided by [10] from censuses collected by National Statistics Offices in each country. Estimates for Brazil, Dominican Republic, Ecuador, Mexico, and Panama refer to year 2010. Estimates for Costa Rica, Trinidad and Tobago, and Uruguay refer to year 2011. Symbols in columns (3), (6) and (9) denote statistical significance:

[*] 0.10,

[†] 0.05, and

[‡] 0.01.

those with motor limitations (23.3 percent and 26.5 percent) and hearing limitations (17.5 percent and 17.8 percent). In the case of persons with vision disabilities, the average employment gap is still positive and large (5.4 percent and 11.7 percent), but smaller than for the other types of disability.

The variability between countries in the employment gap, as in disability prevalence rates and education gaps, may reflect multiple factors (including differences in the definition and recording of disability). Still, looking at the region's best performers' approaches might provide relevant examples and guidance to countries lagging.

## Discussion

The World Health Organization (WHO) provided an estimate of the global prevalence of disability using 2002–2004 surveys in its World Report on Disability [23]. Their estimation calculated a global disability prevalence rate of 15 percent, which became widely used in technical and popular media. This global estimate masks substantial heterogeneity across countries and hinges on the definition and method used to measure disability.

Population estimates of disability prevalence are likely to be affected by the question used to elicit an individual's disability status. Comparisons across countries are best made within age groups, as age structures vary widely and some countries do not report prevalence rates for children younger than certain ages. Comparing prevalence rates for 65–74 year-olds for the eight countries in our study with those in Canada and the U.S. (measured using WG

questions), we find that the countries in our sample that more closely applied the WG questions fall close to the Canadian and U.S. estimates range. In the U.S., the disability prevalence rate among this age group is 24.4 percent, and in Canada it is 32 percent. Brazil has the highest prevalence rate at 62.1 percent, followed by Dominican Republic (45.4 percent) and Uruguay (39 percent). The countries that used a medical approach or filter questions have lower prevalence rates. The estimated disability prevalence among 65 to 74 year-olds in the countries in our sample are: Brazil (2010) 62.1 percent; Dominican Republic (2010) 45.4 percent; Uruguay (2011) 39 percent; Costa Rica (2011) 33.6 percent; Panama (2010) 28.7 percent; Mexico (2010) 23.2 percent; Ecuador (2010) 17.1 percent; Trinidad and Tobago (2011) 12.1 percent. In Africa, overall (whole population) prevalence of disability measured using WG questions (and the same threshold of "some difficulty") was estimated to be 12.85 percent in Ethiopia, 10.78 percent in Malawi, 15.05 percent in Tanzania, and 15.36 percent in Uganda, using surveys collected circa 2010 [24].

The country's demographic structure also affects the overall disability prevalence rate. A country further along with the demographic transition, with an older age structure, would be expected to have a, ceteris paribus, higher overall prevalence rates than a country with a younger age structure.

Using more recent data from censuses and exploiting that some countries have used the preferred functional approach to measure disability, we aim to provide updated estimates of disability prevalence for Latin American and the Caribbean. As the fastest aging region worldwide [25], it is crucial to explore how disability prevalence will change in LAC over the next few decades. We take the disability prevalence rates for each sex and age group (using 5-year intervals until 99 years old, and a group for people 100 years old or older) estimated in Uruguay (2011) as a reference, and extrapolate the overall disability prevalence to other countries in LAC based on their demographic composition. We use the population estimates by United Nations Economic Commission for Latin America and the Caribbean (CEPAL) for 2020 and 2050 [26]. Section 5 ("Estimating Total Population with Disability: Extrapolation") in the technical appendix (S1 Appendix) provides more details about this exercise. We selected this country as a reference for the projections because of its high fidelity to the WG questions to measure disability, using limitations rather than medical conditions and avoiding distortionary filters. In S14 Table, we repeat the exercise using sex and age-specific disability prevalence rates estimated for Dominican Republic (2010). The Dominican Republic (2010) also tracked the WG recommendations closely (despite the use of Yes/No responses), and it is at a different stage in the demographic transition relative to Uruguay The very high prevalence of vision disabilities in Brazil made us reluctant to use Brazil for the projections. The validity of estimations based on this approach relies on three assumptions. In addition to the validity of the population projections by the UN. The first assumption is that questions and categories in the Uruguay (2011) censuses correctly captured the disability status of the surveyed people. The second assumption is that the age and gender-specific rates are constant over time. The last assumption is that, for each demographic group (for example, men in their early sixties), differences in the observed prevalence of disability between countries result only from the variation in the survey questions' approach and interpretation. The validity of this last assumption is not straightforward, as countries in the region are in different stages of economic development. However, using Uruguay as a benchmark has the advantage that, as one of the most developed countries in the region and with the best social protection network, it is likely to provide a lower bound to the incidence of disability for the rest of the region. Regarding the stability of rates over time (second assumption), our estimates are *ceteris paribus* in the sense that they cannot account for any significant reform on inclusion policies or technologies that might be introduced in the region. This exercise shows a simple method of how age and sex-

**Table 6. Estimates for disability prevalence using Uruguay 2011 prevalence rates.**

| | 2020 Population Estimates | | | | | | 2050 Population Estimates | | | | | |
|---|---|---|---|---|---|---|---|---|---|---|---|---|
| | # Persons (000s) | | | Rate (%) | | | # Persons (000s) | | | Rate (%) | | |
| | All | Female | Male | All | Female | Male | All | Female | Male | All | Female | Male |
| Argentina | 6,589 | 3,859 | 2,731 | 15.9 | 18.1 | 13.6 | 9,900 | 5,740 | 4,160 | 19.3 | 21.9 | 16.5 |
| Bahamas | 53.8 | 31 | 22.8 | 14.7 | 16.4 | 12.9 | 86.4 | 50.5 | 35.9 | 19.7 | 22.4 | 16.9 |
| Belize | 43.7 | 23.6 | 20.1 | 12.2 | 13.1 | 11.3 | 92.1 | 52.6 | 39.5 | 17.2 | 19.3 | 15.1 |
| Bolivia | 1,400 | 767 | 633 | 13.3 | 14.7 | 12 | 2,493 | 1,400 | 1,094 | 17 | 18.9 | 15 |
| Brazil | 30,338 | 17,420 | 12,917 | 15.3 | 17.2 | 13.3 | 48,640 | 28,240 | 20,400 | 22.3 | 25.2 | 19.2 |
| Barbados | 53.1 | 30.8 | 22.3 | 19.5 | 21.9 | 16.9 | 66.1 | 37.3 | 28.8 | 25 | 27.8 | 22.1 |
| Chile | 3,012 | 1,732 | 1,280 | 16.8 | 19 | 14.5 | 4,574 | 2,575 | 1,999 | 23.6 | 26.3 | 20.8 |
| Colombia | 7,003 | 3,994 | 3,008 | 14.8 | 16.6 | 13 | 11,497 | 6,581 | 4,916 | 21.6 | 24.3 | 18.8 |
| Costa Rica | 744 | 415 | 330 | 15.7 | 17.4 | 13.9 | 1,268 | 711 | 557 | 23 | 25.7 | 20.4 |
| Dominican R. | 1,364 | 752 | 612 | 13.9 | 15.2 | 12.5 | 2,255 | 1,284 | 971 | 18.8 | 21.1 | 16.4 |
| Ecuador | 2,201 | 1,216 | 985 | 13.8 | 15.2 | 12.4 | 4,087 | 2,282 | 1,805 | 18.7 | 20.9 | 16.6 |
| El Salvador | 843 | 502 | 341 | 14.3 | 15.8 | 12.4 | 1,264 | 779 | 485 | 19.3 | 22.3 | 15.9 |
| Guatemala | 1,857 | 1,038 | 819 | 11.7 | 12.9 | 10.5 | 3,982 | 2,264 | 1,718 | 16 | 18 | 13.9 |
| Guyana | 97.6 | 53.7 | 43.9 | 13.7 | 15.1 | 12.3 | 139 | 77.9 | 60.9 | 18 | 20.3 | 15.7 |
| Honduras | 1,058 | 583 | 475 | 11.9 | 13.1 | 10.7 | 2,214 | 1,241 | 972 | 17.1 | 19.2 | 15 |
| Haiti | 1,215 | 675 | 540 | 12 | 13.1 | 10.8 | 2,122 | 1,204 | 918 | 15.4 | 17.3 | 13.5 |
| Jamaica | 407 | 225 | 182 | 14.9 | 16.3 | 13.4 | 566 | 324 | 242 | 20.2 | 22.6 | 17.7 |
| Mexico | 16,577 | 9,410 | 7,166 | 14.1 | 15.6 | 12.5 | 28,228 | 16,350 | 11,878 | 19.3 | 21.9 | 16.6 |
| Nicaragua | 753 | 428 | 325 | 12.6 | 14.1 | 11.1 | 1,444 | 835 | 609 | 18.1 | 20.6 | 15.5 |
| Panama | 573 | 316 | 257 | 14.6 | 16.1 | 13.1 | 1,069 | 597 | 472 | 19.5 | 21.7 | 17.3 |
| Paraguay | 830 | 446 | 384 | 12.9 | 14.1 | 11.8 | 1,446 | 800 | 645 | 17 | 19 | 15.1 |
| Peru | 4,387 | 2,419 | 1,967 | 14.6 | 15.9 | 13.2 | 7,608 | 4,269 | 3,339 | 20 | 22.2 | 17.8 |
| Suriname | 74.9 | 42 | 32.9 | 14 | 15.8 | 12.3 | 113 | 65.2 | 47.9 | 17.8 | 20.3 | 15.2 |
| Trinidad & T. | 215 | 122 | 93.2 | 16.4 | 18.4 | 14.4 | 281 | 163 | 118 | 22 | 24.8 | 19 |
| Uruguay | 593 | 357 | 235 | 18.3 | 21.3 | 15.1 | 753 | 441 | 312 | 21.9 | 25.1 | 18.5 |
| Venezuela | 3,742 | 2,120 | 1,622 | 14.4 | 15.9 | 12.7 | 6,386 | 3,710 | 2,675 | 18.4 | 20.8 | 15.8 |
| Other Carib. | 2,364 | 1,349 | 1,016 | 19.4 | 21.8 | 16.9 | 2,922 | 1,656 | 1,265 | 26.1 | 29.2 | 22.9 |
| Total LAC | 88,387 | 50,327 | 38,059 | 14.8 | 16.5 | 13 | 145,495 | 83,730 | 61,765 | 20.3 | 22.9 | 17.6 |

Source: authors' estimations based on data provided by [10] from censuses and surveys collected by National Statistics Offices in each country, and population projections by [26]. The "Other Carib." category includes Antigua and Barbuda, Aruba, Cuba, Curacao, Granada, Guadeloupe, Martinica, St. Vincent and the Granadines, and St. Lucia. Estimates consider individuals aged 5 years and older.

specific disability measures from one country can help estimate and project disability prevalence rates in other countries.

We estimate a total population of 88.3 million people with disabilities in the region in 2020 and a total of 145.5 million in 2050. Table 6 shows the estimates for different Latin American and Caribbean countries using prevalence rates from the Uruguay (2011) census S14 Table reports results using sex and age-specific prevalence rates obtained from Dominican Republic (2010) census. Results using Uruguay rates are similar, which is reassuring about the robustness of the exercise as these two countries have different demographic structures.

Using the demographic structure of countries in 2020 and applying the five-year age and sex-specific prevalence rates as measured from the censuses for Uruguay, we project that prevalence rates in countries with the youngest age structures (Guatemala, Haiti, and Honduras) will increase from around 12 percent in 2020 to 15.4–17.1 percent in 2050. For Barbados and

Uruguay, the countries furthest along in the demographic transition, we project an overall prevalence rate of 21.9–25 percent by 2050. The exercise shows a substantial variation across countries in the prevalence of disability, resulting solely from the population's age structure differences.

## Conclusion

This paper provides estimates for disability prevalence for eight countries in Latin America and the Caribbean. We show that prevalence varies across countries. Differences in methods used to estimate these rates and demographic structures, at least, partly explain this variability. We find marked age and socio-demographic gradients for disability. Also, people living with disabilities have lower educational attendance and completion rates, and gaps in labor market inclusion. These results suggest that countries are far from achieving the standards set out in international frameworks and require effective social programs to promote inclusion for children, youth and working age populations. We find that, in many countries, the prevalence of disability is higher for men at younger ages, but the pattern reverses later in life. We use our data to project rates of disability for the whole region and explore the role of rapid rates of population aging in LAC. Our projections suggest that countries need to systematically plan and implement inclusionary policies to accommodate a growing population of people with disabilities in years to come. An essential first step in that direction is to systematically collect comparable information on disability over time and across countries.

The main strength of our disability estimates is that they rely on representative samples from census data. The use of census data gives us several advantages over other administrative datasets and household surveys. First, it allows us to obtain estimates that are representative of the entire population as opposed to estimates obtained from administrative data. Administrative data usually only include information on beneficiaries of social programs, which are a very vulnerable segment of people with disabilities but are not representative of the entire population with disabilities. Second, compared to studies based on household surveys, we can obtain estimates for small sex-age groups, while complying with reporting guidelines and good practices [22]. Our study's main disadvantage is the heterogeneity in the way disability questions are asked across countries, with many of them not following the WG's best practice. This heterogeneity, of course, sets limits to the comparability of estimates across countries and within countries over time. This disadvantage should be a potent reminder of the need to follow accepted best practices as many countries in the region prepare to launch census data collection in the next few years.

## Supporting information

**S1 Table. GDP per capita and population structure.**
(PDF)

**S2 Table. Sampling method by sample.**
(PDF)

**S3 Table. Missing vs. Non-missing observations: Differences in observable characteristics.**
(PDF)

**S4 Table. Surveys' disability-related questions and response categories.**
(PDF)

**S5 Table. Prevalence of disability: Estimates by country and sex (ages 15 and older).**
(PDF)

**S6 Table. Prevalence of severe disability: Estimates by country and sex.**
(PDF)

**S7 Table. Prevalence of disability by country and sex: Estimates by age group.**
(PDF)

**S8 Table. Prevalence of different types of disability by country and sex: Estimates by age group.**
(PDF)

**S9 Table. Prevalence of disability by country and sex: Estimates by household head education level and age group.**
(PDF)

**S10 Table. School attendance and completion rates and gaps: Estimates by sex.**
(PDF)

**S11 Table. School attendance and completion rates and gaps: Estimates by type of disability.**
(PDF)

**S12 Table. Employment rates and disability Gaps: Estimates by sex.**
(PDF)

**S13 Table. Employment rates and disability Gaps: Estimates by type of disability.**
(PDF)

**S14 Table. Estimates and projections of disability prevalence using dominican republic 2010 prevalence rates.**
(PDF)

**S1 Fig. Prevalence of different types of disability by country and sex: Estimates by age group.**
(PDF)

**S1 Appendix. Technical appendix.**
(PDF)

**S1 File.**
(ZIP)

**S1 Data.**
(XLSX)

## Acknowledgments

The authors thank Luisa Baptista de Freitas and Adriana Castillo Castillo, who provided excellent research assistance. The opinions expressed in this publication are those of the authors and do not necessarily reflect the views of the Inter-American Development Bank, its Board of Directors, or the countries they represent.

## Author Contributions

**Conceptualization:** Samuel Berlinski, Suzanne Duryea, Santiago M. Perez-Vincent.

**Data curation:** Samuel Berlinski, Suzanne Duryea, Santiago M. Perez-Vincent.

**Formal analysis:** Suzanne Duryea, Santiago M. Perez-Vincent.

**Methodology:** Samuel Berlinski, Suzanne Duryea, Santiago M. Perez-Vincent.

**Writing – original draft:** Samuel Berlinski, Suzanne Duryea, Santiago M. Perez-Vincent.

**Writing – review & editing:** Samuel Berlinski, Suzanne Duryea, Santiago M. Perez-Vincent.

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
