## [Decision Letter · Decision Letter 0]

16 Feb 2021

PONE-D-20-35261

Prevalence and Correlates of Disability in Latin America and the Caribbean: Evidence from 8 National Censuses

PLOS ONE

Dear Dr. Berlinski

Thank you for submitting your manuscript to PLOS ONE. After careful consideration, we feel that it has merit but does not fully meet PLOS ONE’s publication criteria as it currently stands. Therefore, we invite you to submit a revised version of the manuscript that addresses the points raised during the review process.

Kindly revise addressing all the issues raised by the reviewers especially your justification for using the reference countries identified.

A rebuttal letter that responds to each point raised by the academic editor and reviewer(s). You should upload this letter as a separate file labeled 'Response to Reviewers'.A marked-up copy of your manuscript that highlights changes made to the original version. You should upload this as a separate file labeled 'Revised Manuscript with Track Changes'.An unmarked version of your revised paper without tracked changes. You should upload this as a separate file labeled 'Manuscript

We look forward to receiving your revised manuscript.

Kind regards,

Gerard Hutchinson, MD

Academic Editor

PLOS ONE

Journal Requirements:

2)  Thank you for stating the following in the Financial Disclosure section:

[The 3 authors (Samuel Berlinski, Suzanne Dureya and Santiago Perez-Vincent) are

employees of the Inter-American Development Bank. No specific grants were received

to produce this document. The Inter-American Development Bank, its board of

directors or the countries they represent had no role in study design, data collection

and analysis, decision to publish, or preparation of the manuscript.].   

We note that one or more of the authors are employed by a commercial company: Inter-American Development Bank

a) Please provide an updated Competing Interests Statement declaring this commercial affiliation along with any other relevant declarations relating to employment, consultancy, patents, products in development, or marketed products, etc. 

Please include both an updated Competing Interests Statement in your cover letter. We will change the online submission form on your behalf.

3) We note that you have indicated that data from this study are available upon request. PLOS only allows data to be available upon request if there are legal or ethical restrictions on sharing data publicly. For more information on unacceptable data access restrictions, please see http://journals.plos.org/plosone/s/data-availability#loc-unacceptable-data-access-restrictions.

4) Please ensure that you refer to Figure 1 in your text as, if accepted, production will need this reference to link the reader to the figure.

5) Please include captions for your Supporting Information files at the end of your manuscript, and update any in-text citations to match accordingly. Please see our Supporting Information guidelines for more information: http://journals.plos.org/plosone/s/supporting-information.

Reviewers' comments:

Reviewer's Responses to Questions

**Comments to the Author**

1. Is the manuscript technically sound, and do the data support the conclusions?

Reviewer #1: Partly

Reviewer #2: Yes

2. Has the statistical analysis been performed appropriately and rigorously? 

Reviewer #1: Yes

Reviewer #2: Yes

3. Have the authors made all data underlying the findings in their manuscript fully available?

Reviewer #1: Yes

Reviewer #2: Yes

4. Is the manuscript presented in an intelligible fashion and written in standard English?

Reviewer #1: Yes

Reviewer #2: Yes

5. Review Comments to the Author

Reviewer #1: This is an interesting topic and it has great relevance for LAC. However, before the article can be considered for publication there are several points is will need to improve.

In general, the article is interesting, but it will need that each section includes the necessary information to understand why this research is important, what was done and the results.

Definition of disability: Some of the census have used the Washington Group questions and recommendations, it will be advisable to use those questions and definition when possible.

Paragraph 3 of the introduction should be in the methodology or discussion.

Most of the information included in the introduction is part of the results or discussion.

The authors will need to write a proper introduction, methodology and a results section.

It is not clear why data from Dominican Republic and Uruguay were used to compute specific country projections.

It is recommended that the authors rewrite the introduction as an introduction, for now it is the summary of different results and it does not present the importance of this paper, the gap on the knowledge is not clear or the importance of this study for the LAC region. In addition, it is important to present what information exist on disability, what is the evidence on this topic for LAC and to have a clear discussion on what is the theoretical perspective that the paper uses.

Data and methods:

Although it is really important to present the definitions of disability included in each survey, it is recommended to reduce the length of that section and be more concrete about the definition used and what were the methodological decisions around this topic.

In addition, it is not clear if the harmonized data was used, why you were using the non-harmonized questions on disability? This is not clear in the text.

Also if you presented the recommendation made by the WG, why you did not follow them and instead follow recommendations that are not the ones with the largest support on the disability measurement?

It is important to discuss what are the implications of how you measured/defined disability in the article

It will be advisable to reduce the length of the computation of disability prevalence rates, this section can be shorter and more concrete providing only the information and formulas the reader needs.

It is necessary that all analysis presented in the result section are introduced in the methodology, for example in the results the authors mention we estimate linear regression models for each country using a disability indicator as the dependent variable and a female indicator variable as a dependent variable. But this was never introduced in the methodology.

Statistical significant levels should be presented for all figures

Why you selected to analyse the prevalence only for vision, hearing, motor, and cognitive impairments. This analysis also should be included in the methodology.

Why only socioeconomic characteristics at the hh level were included? Why not to analyse level of education or employment of the person?

Reviewer #2: Manuscript #: PONE-D-20-35261

Title: Prevalence and correlated of disability in Latin America and the Caribbean: Evidence from 8 National Censuses.

General: This manuscript estimated the prevalence rates of disability in 8 countries from Latin America and the Caribbean (LAC) using systematic random samples from recent population censuses for 2010 and 2011. Disability prevalence for 2020 and 2050 were estimated.

Abstract: Concise and specific.

Introduction: Comments:

-Please limit the findings of this study to the results or discussion sections.

-It will be very helpful if in this section to compare the disability status in LAC with other countries in the American Continent.

-Please state in this section that estimates of the disability prevalence will be calculated for 2020 and 2050.

Methods: Comments:

-Please provide a rationale why Dominican Republican and Uruguay were used as reference countries.

-Please provide the missing information on disability for each country.

-How different or similar those participants not included in the analysis with those included.

Results: Comments:

-Please provide in the table or figure title for which year are the prevalence estimates (Table 2, Figure 3 to 6, Table 3 and 4, and Tables A.4 to A.14).

Discussion: Comments:

-This section it will benefit from a comparison with other continents. Where LAC stands in the prevalence of disability worldwide.

-A paragraph of strengths and study limitations is needed.

References: All appropriate.

Minor comment:

-On page 11, 2nd paragraph, line 7, the word of “incidence” should be replaced with “prevalence”.

6. PLOS authors have the option to publish the peer review history of their article (what does this mean?). If published, this will include your full peer review and any attached files.

Reviewer #1: **Yes: **Monica Pinilla-Roncancio

Reviewer #2: No

---

## [Decision Letter · Decision Letter 1]

29 Jun 2021

PONE-D-20-35261R1

Prevalence and Correlates of Disability in Latin America and the Caribbean: Evidence from 8 National Censuses

PLOS ONE

Dear Dr. Berlinski,

Thank you for submitting your manuscript to PLOS ONE. After careful consideration, we feel that it has merit but does not fully meet PLOS ONE’s publication criteria as it currently stands. Therefore, we invite you to submit a revised version of the manuscript that addresses the points raised during the review process.

Regarding the suggestion of one of the referees to reject the publication of the paper, please take into account their comments, as well as those of the others, when considering the new version of the document. 

We look forward to receiving your revised manuscript.

Kind regards,

Maximo Rossi, PhD Economics

Academic Editor

PLOS ONE

Additional Editor Comments (if provided):

Dear Dr. Berlinski,

I inform you that, considering the comments of the referees with which I generally agree, it has been decided to recommend a major revision of the paper. Regarding the suggestion of one of the referees to reject the publication of the paper, please take into account their comments, as well as those of the others, when considering the new version of the document. Sincerely, M. Rossi

Reviewers' comments:

Reviewer's Responses to Questions

**Comments to the Author**

1. If the authors have adequately addressed your comments raised in a previous round of review and you feel that this manuscript is now acceptable for publication, you may indicate that here to bypass the “Comments to the Author” section, enter your conflict of interest statement in the “Confidential to Editor” section, and submit your "Accept" recommendation.

Reviewer #1: (No Response)

Reviewer #3: (No Response)

Reviewer #4: (No Response)

Reviewer #5: (No Response)

2. Is the manuscript technically sound, and do the data support the conclusions?

Reviewer #1: Yes

Reviewer #3: Partly

Reviewer #4: Partly

Reviewer #5: Partly

3. Has the statistical analysis been performed appropriately and rigorously? 

Reviewer #1: Yes

Reviewer #3: Yes

Reviewer #4: Yes

Reviewer #5: Yes

4. Have the authors made all data underlying the findings in their manuscript fully available?

Reviewer #1: Yes

Reviewer #3: No

Reviewer #4: Yes

Reviewer #5: No

5. Is the manuscript presented in an intelligible fashion and written in standard English?

Reviewer #1: Yes

Reviewer #3: Yes

Reviewer #4: Yes

Reviewer #5: Yes

6. Review Comments to the Author

Reviewer #1: Use the income classification of the country and not developing and developed countries.

The introduction improved a lot, but it will benefit from including more information on previous research in the region.

The result section has improved a lot. However, sentences such as the one between lines 214 and 217 confuse the reader. Those sentences should be included in the discussion section.

The WG questions should not be used for individuals younger than 15 years. The prevalences that the authors are presenting for individuals younger than 15 are affected by measurement error, and it is not recommended to estimate the prevalence of disability for this group using these questions.

The discussion needs still a lot of work; in its current state, it only summarizes what the authors did but not discusses the results.

Given the different definitions of disability and that some countries use the WG questions and a severity scale, it is recommended to estimate the prevalence using other categories.

In school attendance, it is essential to acknowledge that school attendance would vary between impairments even if the measure of disability were correct.

Reviewer #3: Prevalence and Correlates of Disability in Latin America and the Caribbean: Evidence from 8 National Censuses

Review

The comments on the revised version of the paper are organized by sections below.

Data and methods – Data sources and definitions

In footnote 2 the authors state that the country selection is driven by data restrictions. It is useful to know which LAC countries that are included in IPUMS do not include disability information (other than Argentina).

I have several doubts about the missing data treatment. First, the missing proportions reported in table 1, refer to individuals with no data on disability, the ones that are not considered in the study? Or it reflects missing information in at least one disability variable? Second, the large proportion of missing data in Uruguay is outstanding. This seems to be caused by age-filters in the questions. As there is no information for children under 2, the authors should consider dropping the 0-2 population from the study (all the countries for comparability). Third, the information on differences between missing and non-missing information presented to reviewer 2 should be mentioned in the paper.

Table S2 show very important differences in the formulation of disability questions. This is the most important limitation of the paper. However, none of the countries follow the WG recommendation (https://www.washingtongroup-disability.com/question-sets/wg-short-set-on-functioning-wg-ss/). The compliance with the WG should be clearly stated by the authors. One issue that is not addressed in the paper is that the WG recommends that the answers to the limitation questions are graded, avoiding YES/NO formulations. The YES/NO formulation could discourage people with mild disability to declare disability, and lead to lower rates. Actually, this could explain (partially) the differences between countries observed in the paper. The two countries that ask for severity (Brazil and Uruguay) are the ones that show higher prevalence, followed by the YES/NO countries. The lowest prevalence occurs in countries that ask directly about disability (this is clearly stated by the authors).

In any case, I suggest that you directly refer in this section which are the countries that better adjust to WG recommendations (the ones used as reference for the projections).

Table S2: Wording of the Surveys' Disability-related Questions and Response Categories:

• In footnote 8 the authors note that Panama includes a filter for the general questions on disability. This is not clear in the S2 Table.

• In the third question for Trinidad and Tobago “Does the long-standing disability prevent (N) [the respondent] from doing any of the following? [Respondents indicate both the activity and its level of diffculty]” the authors do not list the activities.

• Notes on Table S2 should be revised.

Prevalence of Disability: Estimates by country:

The discussion of the prevalence country differences in terms of the formulation of the questions is not complete in my opinion, as it does not consider the severity options in the answers.

Line 284: the oldest group is 81+

Line 296: “fall close to” I think this may be stretching the interpretation (Brazil almost duplicates Canadian figures…). Actually, LAC figures are surprisingly high compared to the other 5 countries. Does this contradicts in any sense the socioeconomic gradient found within countries?

Type of disability: the age groups when analyzing the prevalence by type are somewhat strange. Why do the authors separate children in two groups? Why do they start the elderly group in 56?

The authors mention in the introduction the relevance of studying he association between poverty and disability. However, they study the socio-demographic gradient using the level of education of the household head. I understand that this comes from the lack of income and expenditure measures in the censuses data. This should be noted in the text. Also, the authors should consider trying a different specification for this socio-economic variable (ie. average educational years of the household adults) to explore the robustness of their results.

Disability Gaps:

I was at first stoke by the fact that Trinidad and Tobago had always the larger gaps, followed by Ecuador. These are the countries with lower prevalence (due to the formulation of the questions). The people that are identified as disabled are very different in these countries than in Brazil or Uruguay, they are probably the more severe. This should be at least clarified in the paper. In this sense, the note made in lines 427-432 about better and worse performing countries should be reconsidered.

Projections.

Regarding the selection of the countries on which to base the projections, the WG recommends that the answers to the limitation questions are graded, avoiding YES/NO formulations. I understand that the two countries that follow more closely the recommendation are Brazil and Uruguay. Dominican Republic´s questions allow for YES/NO answers. The authors should reconsider the inclusion of Dominican Republic rates for the projections or justify this based on different arguments. Also, they should justify he exclusion of Brazil.

Assumptions. One central assumption is not mentioned: that age- and gender-specific rates are assumed to be constant in time.

Reviewer #4: Review report for: Prevalence and Correlates of Disability in Latin America and the Caribbean: Evidence from 8 National Censuses

Journal: PLOS

Summary

This paper addresses an interesting research question: What is the level of prevalence of disability in 8 countries from Latin America and the Caribbean?

The question has strong policy relevance and has potentially academic relevance. In particular, evidence on this question could provide important insights to better measure the prevalence of disability in these countries.

The empirical application is based on cross-section data from more recent censuses from LAC countries (2010 or 2011).

The paper concludes that the prevalence varies across countries. Differences in methods used to estimate these rates and demographic structures, at least, partly explain this variability. The results suggest marked age and socio-demographic gradients for disability.

General comments

I find the paper's research question is very relevant, and the data-set available in the study could support a contribution in the field, although there is a limitation regarding the information used in the paper (in terms of comparability). I think that more analysis and discussion are necessary to support the main conclusions.

The paper aims to contribute to two fields of the literature. On the one hand, the paper contributes evidence about the prevalence of disability for eight countries from Latin America and the Caribbean (LAC). On the other hand, the authors suggest a potential methodological contribution regarding the approach (and the instrument) used to measure disability. I found that the paper advance in the first issue, and potentially provides an original contribution. Regarding the second point, more analysis is necessary to provide an original contribution.

Note that, although my evaluation is critical, I appreciate the authors' efforts. The paper addresses an important but understudied question. I find that the work with the data is rigorous and wide. Another merit is the inclusion of 8 different countries and the effort to obtain comparability of results I think this is a promising project, which makes me believe that a revised version of the paper could be well published. I have some suggestions for the improvement of the paper.

Introduction

• I think that the paper could benefit so much if the authors deepen in the motivation, the discussion of their main empirical findings, the policy implications, and the recommendations to better measure prevalence.

• It would be useful to present the main results of the paper in the introduction. Also, it would be necessary to relate them with the main contribution of the paper mentioned in the introduction and with the previous literature. I think that the author may use the introduction to explain how their finding contributes to the existing knowledge.

• I recommend to the authors to present a discussion on what is the theoretical perspective that the research adopts to address this issue. In addition, it is relevant to present the previous evidence on this topic for LAC.

• The paper said (lines 18-19), “We then examine the association between disability and poverty…”. However, this issue is not addressed in the next sections. I understand that the authors have not microdata about household income and poverty levels at the household (or individual) level. To explore that association maybe they could use secondary database (e.g. National Household surveys) and analyze that relationship at regional levels. More in general, the regional heterogeneity within countries is not addressed, and could be interesting to extend the analysis.

• I think that the motivation could improve if the authors justify the novelty of the selected countries. In the current version of the manuscript, the inclusion criteria are based on data availability. The set of countries seems to be heterogeneous, but there is not a systematic discussion about this (demographic, cultural, institutional differences). Maybe we could learn anything about that heterogeneity.

• I think that the motivation could be related to the potential policy implication associated with the main results. Could you explain in more detail the policy implications of your findings? (Maybe this issue could be considered in the conclusion section)

• To advance in the second contribution, it could be useful to incorporate some specific recommendations about disability questions that could be considered in the future census of these countries.

Data and method

• In lines 65 and 66 the text said “For our analysis, we construct a disability variable that maximizes comparability across countries, restricting to the WG questions when available.” Then the Table S2 provides details about the questions to measure disability in each census. It could be useful to include in this Table the WG questions as reference. This could help to describe the synthesis about the comparability across countries.

• When the authors introduce the socioeconomic variables, they could mention when they refer to the household head.

Estimation method

• Is it possible to better link the parameters of equations 1 and 2 with the results about the average prevalence by groups?

• Equation 2 includes cohort fixed effect. However, the note in Table 3 includes “age, fixed effect as dependent variables”. I think that is the same variable, but it could be more appropriate to use a unique label. I identified a similar situation in other parts of the text (e.g. line 174). A minor issue, in the note of Table 3 the authors refer to the fixed effect as a dependent variable. I think that is a typo mistake.

• In this section, the authors could include the reference to Section 5 of the Appendix, entitled “Estimating Total Population with Disability: Extrapolation. A minor issue: In this section, equation 17 refers to the Dominican Republic, but it does not consider Uruguay.

Results

• The first general comment is that this section could improve if the authors advance in the interpretation of the results, establishes some hypotheses, and explore the ways to strengthen dialogue with the previous papers.

• Could you explain the origin of the disability gap by sex? I expect that for these countries the average male life expectancy is lower than females. Is the origin of the sex gap that men with disabilities tend to die younger than the other men or than women with disabilities? Is it that there are more women between adults older which introduces persons with new disabilities? Evidence about these issues could be policy implications.

• Table 2/Figure 1: Brazil shows the highest prevalence. Could you suggest any hypothesis to explain that result?

• Table 3: It could be useful to include in the Table the prevalence per group and country to identify the magnitude of the gap.

• The household head analysis considers the education level. This analysis could be introducing a cohort bias because those people with primary incomplete could be older.

• Table 4. Please explain how this table considers the educational lag of individuals.

• Table 4 It would be useful to comment in this case whether there are differences by gender. Idem comment for Table 5.

• In lines 286 and 287 the text states “In all age groups for practically all countries, the prevalence of disability is significantly higher in households with low education levels.” I not sure that this result is confirmed for persons under six years old,

• In order to interpret the results in Table 4, it would be useful to consider more contextual information. For example, the authors could report whether there are differences in the education supply for children with disabilities.

• Table 5. Why you selected to analyze only the employment rates? Participation and unemployment rate might provide complementary information, and provide new insights for policy implications.

• Line 364 said: “The variability between countries in the employment gap, as in disability prevalence rates and education gaps, may reflect multiple factors (including definition and recording of disability).” This statement is so general that it questions the results previously presented. At this point, as a reader, emerges the question about what we have to learn from the study and what its original contribution is. Moreover, I think that this statement seems contradictory with the comments of the first paragraph in the conclusion section.

Discussion

• Please provide a brief discussion about the validity of the assumptions presented in lines 387 to 391. Note that the paper does not present the differences between countries (for example, in terms of demographic structure).

• Information from the previous census could be useful to assess the validity of the projections and assumptions included in the analysis (It is just an idea; I do not expect an answer on this point.).

• How these projections of people living with a disability may affect the need for resources for inclusion policies? Given that projection, the author could provide any idea about the magnitude of the fiscal effort that would imply maintaining current inclusion policies.

Reviewer #5: This paper analyzes the prevalence of disability in Latin America and the Caribbean (LAC). Based on census microdata (circa 2010) from eight countries, the authors find that overall prevalence of disability is greater among females in six countries, and the association between disability and sex is moderated by age. Their estimates indicate that vision and motor disabilities are the most prevalent, and motor, visual, and hearing disabilities have a strong age gradient. The authors also report that people living with disabilities have lower educational attendance and a lower employment rate. Finally, the paper projects that the number of people with disabilities in LAC will increase from at least 13.7 percent of the population in 2020 to 19.3 percent in 2050.

The paper discusses a relevant topic. There is little information available on the prevalence of disability, and even less that might provide opportunity to make comparisons between countries. This information allows the article to make relevant contributions in these specific aspects. However, the paper has some weaknesses that will be a challenge to overcome. I believe that the empirical analysis is not very robust, firm conclusions are drawn with little evidence, and the work lacks a conceptual framework to help interpret the results. On this last point I would like to point out just one example: a link is established between disability, poverty and inequality, but on what elements/evidence is this link based? To what degree could this relationship be understood as a mechanical phenomenon, or how much is it the product of stigmatization or other individual behaviors? In short, an informative/descriptive contribution is proposed without providing explanations of the phenomenon. While it is helpful to have an overview of the disability situation in LAC, it seems an insufficient contribution for an academic article.

Below, I list some points about the results that I believe can improve the article.

In Table 2

• The prevalence of disability varies widely between countries. How are these differences interpreted in the absence of a conceptual framework? Since an effort is made to reconcile the various indicators used to approach the question of disability, how much does the heterogeneity in the questions contribute to the differences between countries? Is it not possible to decompose these results? Or perhaps it is the demographic structures of the countries that explain these results? I am struck by the prevalence in the case of Brazil, given its age structure.

• Is there an explanation for the absence of significant differences between men and women in Trinidad and Tobago and Mexico?

In Table 3

• The prevalence of disability according to age groups shows different behaviors between countries. The only countries with similar behavior are Trinidad and Tobago with Costa Rica and Brazil with the Dominican Republic. How are these heterogeneous trajectories explained?

• Is it not possible that demographic structure is at play in within the age groups? Wouldn’t the life expectancy of men and women affect the results, in a heterogeneous way, for those over 65 years of age? To observe this, within age groups, you could generate interactions of sex with age.

In Tables 2 and 3

• Couldn't the differences between countries be the consequence of the prevalence of different types of disability in each of the countries? Is it impossible to make the same estimates according to the types of disability (or excluding the types of disabilities one by one)?

In Table 4

• In this case, the type of disability seems particularly relevant. What would happen if cognitive disabilities are excluded from the setting of the dependent variable? This can also apply to Table 3.

• How do employment gaps correlate with educational gaps? In other words, can't employment gaps stem from educational gaps? Again, the absence of a conceptual framework it does not allow interpret these results.

7. PLOS authors have the option to publish the peer review history of their article (what does this mean?). If published, this will include your full peer review and any attached files.

Reviewer #1: No

Reviewer #3: No

Reviewer #4: No

Reviewer #5: No

---

## [Author Response · Author response to Decision Letter 1]

11 Sep 2021

Responses to the editor and reviewers were uploaded.

---

## [Editor Report · Decision Letter 2]

7 Oct 2021

Prevalence and Correlates of Disability in Latin America and the Caribbean: Evidence from 8 National Censuses

PONE-D-20-35261R2

Dear Dr. Berlinski,

We’re pleased to inform you that your manuscript has been judged scientifically suitable for publication and will be formally accepted for publication once it meets all outstanding technical requirements.

Kind regards,

Maximo Rossi, PhD Economics

Academic Editor

PLOS ONE

Additional Editor Comments (optional):

Dear Samuel Berlinski,

the new version of the paper has incorporated most of the comments received. According to the current state of the paper, I recommend its publication. Best. Maximo
---

## [Editor Report · Acceptance letter]

19 Oct 2021

PONE-D-20-35261R2 

Prevalence and Correlates of Disability in Latin America and the Caribbean: Evidence from 8 National Censuses 

Dear Dr. Berlinski:

I'm pleased to inform you that your manuscript has been deemed suitable for publication in PLOS ONE. Congratulations! Your manuscript is now with our production department. 

Kind regards, 

on behalf of

Dr. Maximo Rossi 

Academic Editor

PLOS ONE